Mutation density changes in SARS-CoV-2 are related to the pandemic stage but to a lesser extent in the dominant strain with mutations in spike and RdRp

Eskier Doğa 1 2
Suner Aslı 3
Karakülah Gökhan gokhan.karakulah@deu.edu.tr 1 2
Oktay Yavuz yavuz.oktay@ibg.edu.tr 1 2 4
1 Izmir International Biomedicine and Genome Institute (iBG-İzmir), Dokuz Eylül University , Izmir , Turkey
2 Izmir Biomedicine and Genome Center (IBG) , Izmir , Turkey
3 Department of Biostatistics and Medical Informatics, Faculty of Medicine, Ege University , Izmir , Turkey
4 Faculty of Medicine, Department of Medical Biology, Dokuz Eylül University , Izmir , Turkey
Gillespie Joseph
Electronic publication date: 2020 Aug 19
Publication date: 2020
Volume: 8
Electronic Location ID: e9703
Received 2020 Jun 17; Accepted 2020 Jul 22
Copyright: ©2020 Eskier et al.
Copyright year: 2020
Copyright holder: Eskier et al.
License: This is an open access article distributed under the terms of the Creative Commons Attribution License, which permits unrestricted use, distribution, reproduction and adaptation in any medium and for any purpose provided that it is properly attributed. For attribution, the original author(s), title, publication source (PeerJ) and either DOI or URL of the article must be cited.
License URL: https://creativecommons.org/licenses/by/4.0/

Keywords: SARS-CoV-2, COVID-19, Surface glycoprotein, Spike, RNA-dependent RNA polymerase, RdRp, Mutation density

Funding: Turkish Academy of Sciences Young Investigator Program (TÜBA- GEBİP) Yavuz Oktay is supported by the Turkish Academy of Sciences Young Investigator Program (TÜBA- GEBİP). The funders had no role in study design, data collection and analysis, decision to publish, or preparation of the manuscript.

==============================
Since its emergence in Wuhan, China in late 2019, the origin and evolution of SARS-CoV-2 have been among the most debated issues related to COVID-19. Throughout its spread around the world, the viral genome continued acquiring new mutations and some of them became widespread. Among them, 14408 C>T and 23403 A>G mutations in RdRp and S, respectively, became dominant in Europe and the US, which led to debates regarding their effects on the mutability and transmissibility of the virus. In this study, we aimed to investigate possible differences between time-dependent variation of mutation densities (MDe) of viral strains that carry these two mutations and those that do not. Our analyses at the genome and gene level led to two important findings: First, time-dependent changes in the average MDe of circulating SARS-CoV-2 genomes showed different characteristics before and after the beginning of April, when daily new case numbers started levelling off. Second, this pattern was much delayed or even non-existent for the “mutant” (MT) strain that harbored both 14408 C>T and 23403 A>G mutations. Although these differences were not limited to a few hotspots, it is intriguing that the MDe increase is most evident in two critical genes, S and Orf1ab, which are also the genes that harbor the defining mutations of the MT genotype. The nature of these unexpected relationships warrants further research.

Introduction

COVID-19 (coronavirus disease 2019) is an ongoing pandemic that has been observed in 7,553,182 patients and responsible for 423,349 deaths as of 13 June 2020. It is characterized by respiratory system problems and slow onset fever, and is caused by SARS-CoV-2, a novel betacoronavirus of presumably zoonotic origin. It has first been identified in the Hubei province of China in December 2019, with confirmed human transmission in January 2020 (Chan et al., 2020; Riou & Althaus, 2020), and is currently a global concern. COVID-19 has high transmissibility (D’Arienzo & Coniglio, 2020; Petersen & Gökengin, 2020), a capacity of asymptomatic cases to spread the disease (Wong et al., 2020), and poses a high degree of danger to both vulnerable individuals and the healthcare systems via such widespread and invisible transmission (Liu et al., 2020; Basu, 2020, p.). Therefore, despite the disease having a current mortality rate of < 6%, lower than similar diseases caused by betacoronaviruses such as SARS or MERS (Zhang & Holmes, 2020), a fuller understanding of and cure for the underlying pathogen is a high priority for researchers and clinicians everywhere.

Soon after its spread to Europe and the US, a strain of SARS-CoV-2 with two non-synonymous mutations in the RNA-dependent RNA polymerase (RdRp) and spike (S) proteins, namely 14408 C>T (P323L) and 23403 A>G (D614G), became the dominant form particularly in Europe, and to some extent in the US, as well. What makes these two mutations particularly interesting is the critical functions of the RdRp and S proteins: RdRp is the main protein responsible for replication of the viral RNA, while S mediates the binding of viral particles to human cells that express the angiotensin converting enzyme-2 (ACE2) protein, and thereby facilitates viral entry. Claims of increased transmissibility of this new strain due to spike D614G mutation by Korber et al. (2020) were met with caution, as other factors such as founder effect, drift etc. could be responsible for its dominance. Our previous study suggested that RdRp 14408 C>T mutation is associated with SARS-CoV-2 genome evolution and could even be working synergistically with the 23403 A>G (D614G) mutation (Eskier et al., 2020). Although these and other studies provided suggestive evidence that indeed this new strain could have altered phenotypic characteristics (Lorenzo-Redondo et al., 2020; Wagner et al., 2020), only after mechanistic studies are performed in cell culture and animal models will we be able to know whether these inferences are true (Grubaugh, Hanage & Rasmussen, 2020). In this respect, a series of reports from several labs has lent support to the increased transmissibility hypothesis for viruses carrying the 23403 A>G mutation: pseudotyped lenti- and retro-viruses with the mutant spike protein infect human cells in culture more efficiently, compared to those with the wild-type spike (Zhang et al., 2020; Ozono et al., 2020; Yurkovetskiy et al., 2020; Hu et al., 2020; Daniloski et al., 2020). However, the underlying mechanism is less clear: Zhang et al. (2020) suggested increased stability of the spike protein to explain increased infectivity of the mutant viruses, while Daniloski et al. (2020) suggested the mutant protein is more resistant to cleavage. Other studies also pointed to the importance of the spike protein for the unique properties of the SARS-CoV-2 (Walls et al., 2020; Cheng et al., 2020). However, animal studies will be required to further test the effects of the D614G mutation to reach a more definitive conclusion. Nevertheless, many lines of evidence support increased transmissibility hypothesis and therefore, spike mutations should probably be monitored closely and analyzed in-depth.

In our previous study, where we analyzed 11,208 SARS-CoV-2 genome sequences, we showed that RdRp mutations, particularly the 14408 C>T mutation, were associated with SARS-CoV-2 genome evolution and higher mutation density (Eskier et al., 2020). In the current study, we analyzed the time-dependent changes in the mutation densities of several SARS-CoV-2 genes and asked whether the patterns of change were different between the strain with the 14408 C>T / 23403 A>G mutations compared to those without either mutation.

Materials and Methods

Genome sequence filtering, retrieval, and preprocessing

SARS-CoV-2 isolate genome sequences were obtained from the GISAID EpiCoV database and their variants were annotated as previously described (GISAID Initiative, 2020; Eskier et al., 2020). Briefly, the sequences obtained on 24 May 2020 were filtered for full length genomes with high coverage and to remove environmental or non-human host samples. The remaining genomes were edited to mask low-quality base calls and aligned against the reference SARS-CoV-2 genome using the MAFFT multiple sequence alignment program (Katoh & Standley, 2013). Variant sites were retrieved using snp-sites, and their effect on peptide sequences were annotated using the ANNOVAR suite of tools (Wang, Li, Hakonarson, 2010). Due to the low quality and gap-heavy content in a majority of isolates, the 5’ untranslated regions (1–265) and the 100 nucleotides at the 3’ end were also removed from analysis. In addition, we removed isolate genomes without well-defined time of sequencing or geographical location data, to ensure the time and location variables can be clearly associated with the mutations, for a final count of 19,705 genomes.

Mutation density calculation

The annotated variants were separated into synonymous and nonsynonymous groups, with variants in nucleotides not located in gene loci considered synonymous. Gene mutation densities were calculated separately for synonymous and nonsynonymous mutations, by dividing the number of variant sites in the gene locus of an isolate by the length of the locus. Densities for each full open reading frame, including the surface glycoprotein (S), envelope glycoprotein (E), nucleocapsid phosphoprotein (N), and membrane glycoprotein (M) genes were calculated, as well as the RdRp coding region of ORF1ab and the whole genome.

Statistical analysis

The categorical variables were analyzed with frequency tables and descriptive statistics were used for the continuous variables. Shapiro–Wilk normality test was used to examine whether the numeric values had a normal distribution in groups. When the numeric values were not normally distributed, Mann–Whitney U test was used to compare the median values of two independent groups. The level of significance was chosen as 0.05 in all hypothesis tests. IBM SPSS Version 25.0 statistical package was used for statistical analyses.

Results and Discussion

14408 C>T and 23403 A>G mutations are associated with mutation density increase over time

In order to determine whether the SARS-CoV-2 strain with 14408 C>T / 23403 A>G mutations was different from those that carried neither mutation, with respect to time-dependent changes in mutation density, we first determined the number of synonymous and non-synonymous mutations. The number of sites carrying mutations, after the low-quality filters were applied to isolates and the 5′ and 3′ ends, was 7, 870, with 16, 563 potential variants. 11, 095 of these variants were characterized as synonymous, as they did not alter any known peptide sequence, while 5, 468 of them were non-synonymous, as they either changed amino acid residues in the translated proteins, or altered the protein length by changing start or stop codon locations. To eliminate the bias that would be caused by the 14408 C>T and 23403 A>G mutations on the nonsynonymous mutation density of isolates they were present in, which would be substantial considering their high frequency, these two nucleotides were also masked during calculations of non-synonymous mutation density.

For further analysis, we focused on the two countries with the highest number of isolates sequenced, the United Kingdom (UK) and the United States (US), which in total contributed 58.76% of all isolates that passed the quality filters. Furthermore, we focused only on isolates that were sequenced after the first genome with both 14408 C>T and 23403 A>G mutations were identified in the respective geographic regions (26 February 2020 for UK, 28 February 2020 for US). Isolates that carried both of the mutations were named “mutant”, or “MT” in short, and those that were wild-type at both positions were named “wild-type”, or: “WT in short; few isolates that carried only one of the mutations was excluded. In total, the isolates were separated into four categories: UK-WT (1,708 genomes), UK-MT (4,903 genomes), US-WT (1,683 genomes), and US-MT (3,407 genomes). There was a total of 6, 163 variants found in isolates in these categories, with 3,533 of them being nonsynonymous, across 5,693 polymorphic sites.

We calculated the “average mutation density per day per genome” (hereafter referred as mutation density, or MDe for short) for both synonymous and non-synonymous mutations in all four categories, as well as the correlation of MDe with time, using Spearman correlation (Fig. 1). We identified a strong positive correlation between non-synonymous MDe and time in both UK and US in MT samples (ρ = 0.70, p-value < 0.001), however, a much weaker correlation was observed in WT samples ρ = 0.27, p-value (= 0.002), indicating a potential relationship between non-synonymous mutations and 14408 C>T / 23403A>G genotype. A similar correlation was identified for synonymous mutations in MT samples, albeit weaker than for non-synonymous mutations (ρ = 0.64, p-value < 0.001). No significant correlation was identified for synonymous mutations in WT samples (ρ = 0.15, p-value = 0.09). One possible explanation for this tight correlation is that MT genomes accumulate both synonymous and non-synonymous mutations over time, whereas WT strains show more variation in their mutation accumulation rate.

Figure 1 Mutation density has a strong correlation with time in isolates carrying both 14408 C >T and 23403 A >G mutations.

(A-B) Distribution of average synonymous mutations per day in (A) isolates carrying the reference nucleotide for both 14408 and 23403 (WT), and (B) isolates carrying the mutations of interest at both sites (MT). (C-D) Distribution of non-synonymous mutations in (C) WT isolates, and (D) MT isolates. Red dots indicate average density of samples from the UK, blue dots indicate the average density of US samples. The correlation coefficients are calculated using Spearman correlation using the combined sets of both countries.

However, closer inspection of the plots points to further differences between MT and WT SARS-CoV-2 genomes: Rather than a monotonic change in MDe over time, certain time points seem to be critical milestones at which the mutation densities take sharp turns. One of them is roughly Day 100, which is defined as the 100th day following the first SARS-CoV-2 genome sequenced, and corresponds to 2 April 2020 (Fig. 1). Whereas the MDe of WT strains keep increasing until Day 100, for both synonymous and non-synonymous, this trend is disrupted from there on and actually a slight decrease is observed. On the other hand, albeit a slight reduction in the strength of correlation, there is almost no change in the rate of increase in MDe (both synonymous and non-synonymous) for the MT genomes, only a slight decrease after Day 130 is observed. Of note, the synonymous MDe of MT genomes starts increasing after Day 80, compared to the non-synonymous MDe, which shows a steadier increase starting from the beginning.

The next obvious question is, “Why are there such sharp changes in MDe at certain timepoints, particularly around Day 100 for WT and Day 80 for MT?” Even though a simple answer is unlikely, we looked into epidemiological data for clues. Interestingly, Day 80 (13 March 2020) corresponds to the beginning of a 20-day period when daily case numbers started increasing dramatically in both the UK and the US, and is also the day “national emergency” was declared in the US; Day 100 (2 April 2020) roughly corresponds to the end of this period (Dong, Du & Gardner, 2020) (Fig. S1) and the daily case numbers show a slow but steady decline after Day 100 in both countries. It makes sense that both strains accumulated more and more mutations as they spread rapidly between Days 80-100. However, it is less clear why this trend continued only in the MT strain after Day 100, when the strict control measures slowed the spreading of SARS-CoV-2.

Next, we determined how MT and WT strains differ at gene level, in terms of MDe. To do so, we examined the overall distribution of mutations combined for the previously specified time period and compared them between the MT and WT genomes (Fig. S2–S3, Table S1). Only 2 (RdRp, N) of the 12 genes showed consistent differential synonymous MDe in the MT genotype compared to the WT genotype, for UK and US. RdRp MDe was lower in the MT genotype (Mean Rank: 104.91 vs. 40.30, p < 0.001 in UK; 80.26 vs. 52.38, p-value < 0.001 in US), whereas MDe of N gene was higher in the MT genotype (Mean Rank: 37.21 vs. 101.59, p-value < 0.001 in UK; 54.05 vs. 77.41, p-value < 0.001 in US). Although differences were significant in the S gene, they were in the opposite directions, higher in UK (Mean Rank: 61.13 vs. 79.93, p = 0.006) and lower in US (Mean Rank: 81.17 vs. 51.51, p-value < 0.001), for the MT genotype. A closer look pointed to country-specific differences: while only 3 of 12 genes showed differences in non-synonymous MDe between MT and WT genotypes for UK, this number was much higher for US, where 9 of 12 genes showed differential non-synonymous MDe. This marked difference between the two countries is likely due to the differences in the composition of the WT strains, which has much higher variety in the US compared to UK owing to heavier influence of Asian SARS-CoV-2 genomes and multiple independent lines of viral spread around the US (Worobey et al., 2020; Deng et al., 2020). Indeed, such differences are expected, even in the absence of multiple founders, and justify our approach of comparing results of two countries to reach a consensus.

As non-synonymous mutations are subject to stronger selection than synonymous mutations and could have different patterns, we also determined possible differences in their MDe values between MT and WT SARS-CoV-2 genomes. Only 2 (S, N) of 12 genes showed consistent differences between the two genotypes in both countries (Table S1). MDe values in both the S (Mean Rank: 48.88 vs. 91.03, p-value < 0.001 in UK; 58.95 vs. 72.73, p = 0.035 in US) and N genes (Mean Rank: 34.45 vs. 104.09, p-value < 0.001 in UK; 55.52 vs. 76.01, p = 0.002 in US) were higher in the MT genotype. 3 genes (Orf1ab, Orf3a, Orf6) also showed differences in both countries, however, in the opposite directions. In contrast to the synonymous mutations, higher number of genes had significant differences between non-synonymous MDe values in UK, compared to US (4 vs 2; RdRp, M, E, Orf7b vs Orf8, Orf10); interestingly, MDe values of the MT genotype were higher in the first four and lower in the last two, compared to WT. Overall, MDe of MT and WT genotypes at gene level showed gene-specific and country-specific differences, while only N gene had consistent differences for both synonymous and non-synonymous mutations in both UK and US.

Non-synonymous mutation densities increase in later samples

After characterization of MDe for the time period that MT genotype existed in the UK or US, we next tested whether genome-level differences between MT and WT genotypes in terms of time dependent changes in MDe are also present at gene level. Therefore, we aimed to understand whether those differences at genome level are merely due to possible founder effects, or due to differences between MT and WT strains’ accumulation of mutations over time. Since Day 100 appeared to be a turning point, we divided the SARS-CoV-2 genomes into two groups by the time of isolation: early (days 60–100) and late periods (days 101–140), and examined the relationships between the average MDe of the two groups for each gene and country (Figs. 2–3, Table S2). While none of the 12 genes in the WT genotype showed any significant difference in MDe between the early and late periods consistently between UK and US, it was the opposite for the MT genotype: S, RdRp, M, Orf1ab, and Orf3a genes showed increased MDe in the late period compared to the early period. It was particularly evident for the S and Orf1ab genes, as both synonymous and non-synonymous MDe in both genes were significantly increased in the late period in both countries. Interestingly, M gene showed a consistent decrease in the non-synonymous MDe only in the MT genomes, against the common trend of increased MDe in this genotype; however, no such association was observed for the synonymous MDe, it was even the opposite in the UK-MT isolates. On the other hand, the RdRp non-synonymous MDe was increased significantly in the late period in UK-MT isolates, and trended towards increase in US-MT isolates (Mean Rank: 29.76 vs. 38.64, p-value = 0.058); RdRp non-synonymous MDe was also increased in late period UK-MT isolates, however, there was no such increase in the US isolates, consistent with time-independent MDe values (Fig. S2), and with the marginally insignificant increase in synonymous MDe in the same isolates. Among changes that were significant for a single-country, there was an overall tendency towards increased MDe in MT genomes, but the opposite was true for WT genomes: 12/14 of single-country changes in MT genomes were associated with increased MDe, whereas 12/16 of such changes in WT genomes were associated with decreased MDe in different genes. These findings suggest that the steady increase in MDe specific to MT isolates even after Day 100 cannot simply be explained by founder effects, or linkage. One could speculate that at least some of the WT strains accumulated mutations that reduced their fitness under lock-down conditions and therefore led to their elimination in the late period (101–140 days), whereas the MT strain was fit enough to keep most or all mutations and even add new ones in this period. However, almost equal contribution of synonymous and non-synonymous mutations may point to other factors in addition to possible differences in fitness. Although S and Orf1ab genes seem to be the main drivers of the observed genome-wide increase the MDe of MT genomes, it seems that other genes are also contributing to this trend.

Figure 2 Synonymous mutation density rates do not shift significantly between early and late periods when comparing WT and MT samples.

(A-L) Synonymous mutation density of regions of interest UK-WT, UK-MT, US-WT, and US-MT samples divided into early period (60–100 days) and late period (101–140 days), with gray bars indicating early period samples, and orange bars indicating late period samples for (A) S, (B) E, (C) M, (D) N, (E) RdRp, (F) Orf1ab, (G) Orf3a, (H) Orf6, (I) Orf7a, (J) Orf7b, (K) Orf8, and (L) Orf10 regions. The average mutation density per day is calculated in average nucleotides mutated per gene/locus across all isolates. Mutations in the RdRp coding region are included in both the RdRp plot and the Orf1ab plot.

Figure 3 MT samples display higher non-synonymous mutation density increases in late period compared to WT samples.

(A-L) Non-synonymous mutation density of regions of interest in UK-WT, UK-MT, US-WT, and US-MT samples divided into early period (60–100 days) and late period (101–140 days). Gray bars indicate early period samples, and orange bars indicate late period samples for (A) S, (B) E, (C) M, (D) N, (E) RdRp, (F) Orf1ab, (G) Orf3a, (H) Orf6, (I) Orf7a, (J) Orf7b, (K) Orf8, and (L) Orf10 regions. The average mutation density per day is calculated in average nucleotides mutated per gene/locus across all isolates. Mutations in the RdRp coding regions are included in both the RdRp plot and the Orf1ab plot.

Figure 4 Mutation densities in MT isolates increase more rapidly compared to WT isolates after first identified MT genome.

(A-D) Barplots for isolates in the 60–80 day time period displaying (A) synonymous mutations in RdRP, (B) synonymous mutations in S, (C) non-synonymous mutations in RdRp, and (D) non-synonymous mutations in S. (E-H) Barplots for isolates in the 121–140 day time period displaying (E) synonymous mutations in RdRP, (F) synonymous mutations in S, (G) non-synonymous mutations in RdRp, and (H) non-synonymous mutations in S. The barplots show 100 nucleotide-long bins indicating regions on the genome in the x-axis, and the total number of isolates with mutations in the indicated regions in the y-axis. Green bars indicate WT isolates and orange bars indicate MT isolates.

Isolates with 14408 C>T & 23403 A>G mutations (MT genotype) show steady increase in mutations in both S and RdRp over time

To further investigate whether the shift in MDe happens due to a few distinct sites that are linked with the MT mutations, and could possibly directly affect the host adaptation process, or whether it is caused by a variety of mutations that spread widely both temporally and spatially in the genome, we further divided the early and late time periods into a total of four periods (60–80, 81–100, 101–120, 121–140), and analyzed the distributions of mutations in the S and RdRp coding regions in these four time periods in a pool containing both UK and US isolates (Fig. 4, Figs. S4–S5). To do so, we calculated the number of mutations in 100-nucleotide bins per isolate sharing a genotype (MT or WT) in that each time period, and observed their distributions, divided into two groups as synonymous and non-synonymous mutations. For the periods 60–80, 81–100, 101–120, and 121–140, the MT/WT distributions were 515/622, 4,044/2,105, 3,119/610, and 632/54, respectively. We can see that in the 60–80 day period, the distribution of mutations across the regions were largely comparable between WT and MT genomes, with the exception of the synonymous 14805 C>T mutation in the RdRp, which is exclusive to WT isolates (Fig. 4A). The ratio of mutations in MT to WT steadily increase over time in both synonymous and non-synonymous categories across UK and US isolates (Figs. S4–S5), with MT genome mutations more visible in most bins in the 121–140 day period, with the major exception of the 14805 C>T mutation being preserved in WT isolates (Fig. 4B). 100-nucleotide binned totals and the mutated isolate ratios for all sites in the regions are available in Supplementary Files 1 and 2, respectively. These observations suggest that the increase in MDe in MT isolates is due to overall increased number of nucleotides in the genome becoming mutated over time, rather than increased frequency of certain genotypes with higher MDe. Although it is unclear whether the 14805 C>T synonymous mutation has any stabilizing effect on the WT MDe, as the mutation is found in only 1,698 samples out of all isolates that passed our quality filters, including non-UK and non-US isolates, and 1,106 of those isolates are UK-WT and US-WT isolates, its high frequency in WT isolates is more likely to be the result of a founder effect, rather than any selection.

Conclusions

Earlier studies of ours, as well as others, pointed at the emergence of RdRp and S mutations and their spread across the world as the pandemic progressed (Pachetti et al., 2020; Eskier et al., 2020). These reports provided evidence supporting possible effects of dominant mutations on SARS-CoV-2 genome evolution. In this study, we reveal an unexpected relationship between the SARS-CoV-2 mutation densities and viral transmission dynamics at population level in humans. While the average mutation densities increased steadily by time during the fast-spreading period of COVID-19 in both countries, this trend ended around Day 100 (2 April 2020), when daily new case numbers started reaching a plateau. It seems, however, that not all strains of SARS-CoV-2 are affected equally by the epidemiological factors (i.e., counter-measures that led to reduced daily new case numbers in UK and US, host factors, treatment regimens, etc.), in terms of mutation densities. SARS-CoV-2 strain with both RdRp 14408 C> T and S 23403 A>G mutations, which became dominant in Europe and US first, differed and continued accumulating both synonymous and non-synonymous mutations after Day 100, particularly in the S and Orf1a genes. Therefore, two related questions remain to be answered: (1) Is there any causal relationship between mutation density changes and viral transmission in population? If the answer is ‘yes’, what is the nature of this relationship? (2) What is the biological basis of differences between the ‘mutant’ and ‘wild-type’ strains, if there is any? It is intriguing that the main drivers of the genome-level increase in MDe were two critical genes, S and Orf1a, which are also the genes that harbor the defining mutations of the MT genotype.

It should also be noted that the MDe of the MT genomes started showing signs of decline after Day 130 (2 May 2020), although no obvious change in the epidemiological data is seen around this date. Further studies are warranted that will investigate the relationship between mutations and the clinical phenotypes, as more data will become widely available.

In conclusion, we propose that monitoring the average mutation densities over time and relating to the epidemiological and clinical data may help establish genotype/phenotype relations, and possibly predict the increases and decreases in new cases.

Supplemental Information

Supplemental Information 1 Supplementary Figures

Click here for additional data file.

Supplemental Information 2 Supplementary Tables

Click here for additional data file.

Supplemental Information 3 Binned distributions of mutations in RdRp and S regions categorized by phenotype and day of sequencing

Position indicates initial nucleotide of 100-nt long bin. Counts indicate average count of mutations in bin in phenotype/days combination. MT indicates presence of both 14408C >T and 23403A >G mutations, WT indicates reference nucleotides.

Click here for additional data file.

Supplemental Information 4 Average number of mutations per genome in isolate phenotype and date for each mutated nucleotide in RdRP and S regions

Mutation counts indicates total count of mutations in nucleotide in phenotype/days combination. MT indicates presence of both 14408C >T and 23403A >G mutations, WT indicates reference nucleotides.

Click here for additional data file.

The authors would like to thank Mr. Alirıza Arıbaş from Izmir Biomedicine and Genome Center for his technical assistance. The authors also would like to extend their thanks to Izmir Biomedicine and Genome Center (IBG) COVID19 platform IBG-COVID19 for their support in implementing the study and the Scientific and Technological Research Council of Turkey (TUBITAK) for their financial support of IBG-COVID19.

Additional Information and Declarations

Competing Interests

Author Contributions

Data Availability

Aslı Suner and Gökhan Karakülah are Academic Editors at PeerJ.

Doğa Eskier, Aslı Suner, Gökhan Karakülah and Yavuz Oktay conceived and designed the experiments, performed the experiments, analyzed the data, prepared figures and/or tables, authored or reviewed drafts of the paper, and approved the final draft.

The following information was supplied regarding data availability:

The processed genotyping VCF file is available at Mendeley Data: Eskier, Doğa; Karakülah, Gökhan; Suner, Aslı; Oktay, Yavuz (2020), “SARS-CoV-2 GISAID isolates (2020 - 05 - 24) genotyping VCF by mutation”, Mendeley Data, v1

DOI 10.17632/jv87xwj7fv.1

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
