# Peer review of "Mutation density changes in SARS-CoV-2 are related to the pandemic stage but to a lesser extent in the dominant strain with mutations in spike and RdRp"

_PeerJ, doi:10.7717/peerj.9703_

## Round 0.1 · original submission · Minor Revisions

Dear Dr. Eskier and colleagues:

Thanks for submitting your manuscript to PeerJ. I have now received three independent reviews of your work, and as you will see, the reviewers raised some minor concerns about the research. Despite this, these reviewers are optimistic about your work and the potential impact it will have on research studying the dynamics of COVID-19. Thus, I encourage you to revise your manuscript, accordingly, taking into account all of the concerns raised by the reviewers.

I look forward to seeing your revision, and thanks again for submitting your work to PeerJ.

Good luck with your revision,

-joe

·

Basic reporting

The authors describe the importance of the mutation density of SARS-COV-2 in the current pandemic related to RdRp and the Spike gene. These are related to the evolution of virus variants according to mutation density. The description of this molecular mechanism in mutations analyzed is essential to understand how SARS-CoV-2 is spreading around the world and opening perspectives on possible drug developments used. The manuscript is mostly well written and referenced. The professional language in English is consistent, as are the results presented through graphs and tables. The hypothesis discussed is in accordance with the data presented and with the literature provided in the manuscript.

Experimental design

The research data is within the scope of the paper and well defined in the materials and methods. The research theme is very actual, relevant, and interesting, also described with sufficient details for understanding and replication.

Validity of the findings

It seems to me that the authors performed a robust analysis and that their fundamental conclusions are consistently based on the generated results. So, there is no need to repeat the experiments, being well analyzed according to the existing literature. The conclusions are in line with the discussion and linked to the central research of the work.

Additional comments

I consider that the manuscript entitled “Mutation density changes in SARS-CoV-2 are related to the pandemic stage but to a lesser extent in the dominant strain with mutations in spike and RdRp ” contains important and relevant new information, which will have impact on the understating of SARS-CoV-2 worldwide spreading and adapting. Despite a scarce literature, it is noted the importance of genomic studies about COVID-19 virus.
The authors made good and extensive use of public available sequence data on SARS-CoV-2, constructed an interesting hypothesis on the relationship between occurrence of RdRp and Spike mutations density.

Reviewer 2 ·

Basic reporting

In their work, Eskier et al. presented a study based on changes in the density of mutations in sars-cov-2. The objective of this work was to understand possible differences between the time-dependent variations of the mutation densities located in the RdRp and S protein.

In the introduction, I believe it is interesting to point out where these two mutations to which the work refers, are located and related to which activities. P323L in the RdRp which is involved in replication of the viral genome and the change of D614G in the Spike glycoprotein which is essential for the entry of the virus in the host cell by binding to the recipient ACE2.

Regarding the D614G change in protein S, which seems to make the virus more infectious, it is important to make it clear that the experiments carried out so far have not been carried out on human cells and that the increase in infectivity should be considered with caution, especially when related this with the increase in transmission that occurs in humans.

Experimental design

No comment

Validity of the findings

About the results, it could have been better discussed with reference to articles that have already demonstrated the relationship between these mutations. For example, In the article, "The hot spots of the emerging SARS-CoV-2 mutation include a new variant of RNA-dependent polymerase" (Pachetti et al.,), the authors showed that new viral variants were spreading across countries.

Reviewer 3 ·

Basic reporting

No comment

Experimental design

No comment

Validity of the findings

No comment

Additional comments

Eskier and coworkes performed an analysis of mutation density (MDe) in SARS-CoV2 genomes over time, showing time-dependent changes related to pandemic stage. In their analyses, they compared the MDe dynamics of synonymous and non-synonymous substitutions in both "wild-type" (WT) and "mutant" (MT, RdRp 14408 C>T and S 23403 A>G) SARS-CoV2 sampled in two different countries (US and UK). MDe where analyzed temporally (in terms of pandemic stage) and spatially (in terms of genome distribution).

Overall I find the study interesting. The paper is well organized. Results are clearly presented and discussed. I have only minor concerns.

1- The authors clearly described and justified the criteria for the SARS-CoV2 genome selection. According to these criteria they reached a dataset of 19705 genomes. In the first paragraph of Results&Discussion is described the number of sites and variants of this dataset, but all the analyses presented in the work are done on a subset of this original dataset, which represents just over half of it (58.76%, including sample from UK and US). I strongly suggest to briefly clarify this point and to deepen the description of this subset (i.e.: sample size for each category; number of polymorphic sites and of potential variants, etc...). Are sample sizes of the two groups (UK and US) comparable? Again, how many genomes constitute each of the four categories analyzed (UK-WT, UK-MT, US-WT and US-MT). Which is the relative proportion of MT and WT samples in the four period analyzed? Please add some comments.

2- Where the analyses of mutation distribution for S and RdRp performed using UK and US genomes jointly? Please clarify. Which is the frequency of 14805 C>T synomymous mutation in the dataset? (Please add the value in the text).



Minor remarks:

I thank you for providing all the raw data, but I think that add also the GISAID EpiCoV database accession date in the main text could be useful for future readers.

Some references are missing in the Introduction Section:
-line 49:
"Therefore, despite the disease having a current mortality rate of <6%, lower than similar diseases caused by betacoronaviruses such as SARS or MERS, a fuller understanding of and cure for the underlying pathogen is a high priority for researchers and clinicians everywhere".
-line 60:
"Although these and other studies provided suggestive evidence that indeed this new strain could have altered phenotypic characteristics, only after mechanistic studies are performed in cell culture and animal models, we will be able to know whether these inferences are true".

line 193: MDE -> MDe

---

## Round 0.2 · accepted · Accept

Dear Dr. Eskier and colleagues:

Thanks for revising your manuscript based on the concerns raised by the reviewers. I now believe that your manuscript is suitable for publication. Congratulations! I look forward to seeing this work in print, and I anticipate it being an important resource for groups studying COVID-19. Thanks again for choosing PeerJ to publish such important work.

Best,

-joe